# DNA Polymerase Delta Exhibits Altered Catalytic Properties on Lysine Acetylation

**DOI:** 10.3390/genes14040774

**Published:** 2023-03-23

**Authors:** Catherine Njeri, Sharon Pepenella, Tripthi Battapadi, Robert A. Bambara, Lata Balakrishnan

**Affiliations:** 1Department of Biology, Indiana University Purdue University Indianapolis, Indianapolis, IN 46202, USA; 2Department of Microbiology and Immunology, University of Rochester, School of Medicine and Dentistry, Rochester, NY 14642, USA; 3Indiana University Comprehensive Cancer Center, Indiana University School of Medicine, Indianapolis, IN 46202, USA

**Keywords:** lagging strand replication, Okazaki fragment maturation, pol δ, lysine acetylation

## Abstract

DNA polymerase delta is the primary polymerase that is involved in undamaged nuclear lagging strand DNA replication. Our mass-spectroscopic analysis has revealed that the human DNA polymerase δ is acetylated on subunits p125, p68, and p12. Using substrates that simulate Okazaki fragment intermediates, we studied alterations in the catalytic properties of acetylated polymerase and compared it to the unmodified form. The current data show that the acetylated form of human pol δ displays a higher polymerization activity compared to the unmodified form of the enzyme. Additionally, acetylation enhances the ability of the polymerase to resolve complex structures such as G-quadruplexes and other secondary structures that might be present on the template strand. More importantly, the ability of pol δ to displace a downstream DNA fragment is enhanced upon acetylation. Our current results suggest that acetylation has a profound effect on the activity of pol δ and supports the hypothesis that acetylation may promote higher-fidelity DNA replication.

## 1. Introduction

Accurate DNA replication is fundamental to the stability of the eukaryotic genome. The efficiency and precision of DNA replication are facilitated by the coordinated action of three main polymerases: alpha (pol α), delta (pol δ), and epsilon (pol ε) [1]. Biochemical analyses have proposed unique roles for each of these polymerases at the nuclear DNA replication fork [2,3]. The leading strand of the DNA double helix is synthesized continuously by pol ε, while the lagging strand is synthesized in a discontinuous manner, via the production of short Okazaki fragments. In mammalian cells, these fragments, ~100–150 nucleotides (nts), are initiated by the primase activity of pol α by the synthesis of an ~8–10 nt RNA primer. The RNA primer is then extended by the polymerase activity of pol α with ~20–22 nts of DNA [4]. Due to its limited processivity and the absence of an inherent 3′ exonuclease activity, pol α is not ideal for accurately and efficiently synthesizing longer chains of DNA [5]. A switching of polymerases takes place, which is mediated by the clamp loading protein, replication factor C (RFC), in which pol α is dissociated and replaced by pol δ. The proliferating cell nuclear antigen (PCNA) clamp encircles the primer-template complex and interacts favorably with pol δ, anchoring the polymerase to the primer terminus. PCNA slides freely along the DNA template, tethering pol δ to the DNA, which consequently increases its processivity and improves the efficiency of the entire replication process [6].

In order to maintain the genome integrity, the RNA/DNA primers produced by the error-prone pol α are removed during the Okazaki fragment maturation. There are two distinct pathways, the short and the long flap pathway, that have been proposed through which pol α synthesized DNA segments are removed [7]. In the short flap pathway, pol δ completes fragment synthesis and runs into the RNA primer of the preceding Okazaki fragment [4]. Upon encountering the 5′ end of the downstream Okazaki fragment, pol δ performs strand displacement synthesis, generating a short 5′ flap, which is cleaved at the base by flap endonuclease 1 (FEN1). The process of strand displacement by pol δ, and flap cleavage by FEN1, is repeated until the pol α synthesized segment has been removed, generating a nicked product that is sealed by DNA ligase I (LigI) [8]. Intermittently, the 5′ flaps escape FEN1 cleavage, and become long enough to be stably bound by the single-strand DNA-binding protein, replication protein A (RPA). The long flaps coated with RPA are refractory to cleavage by FEN1. A second nuclease, Dna2, comes into play, and displaces RPA from the flap, cleaving the flap repeatedly, resulting in a short terminal product that is further processed by the combined action of FEN1 and LigI [9]. 

Despite pol δ being a high-fidelity polymerase, occasional mutations arising from the replication process do occur, resulting in genomic instability and disease conditions. Studies employing ribo-sequencing analyses have shown that the DNA segment synthesized by the error-prone pol α is retained in vivo and constitutes up to 1.5% of the mature genome [10]. Additionally, it has been reported that around three million ribonucleotides are incorporated into the mammalian nuclear genome during replication by the combined action of the three main polymerases [11].

Human DNA polymerase δ holoenzyme is a hetero-tetrameric protein consisting of subunits p125, p50, p66, and p12 [12]. The 125 kDa subunit of the enzyme represents the catalytic subunit and harbors both the polymerase and the 3′–5′ exonuclease activity of the polymerase. All the four subunits have been reported to interact with PCNA [13]. Subunits p66 and p12 have been shown to be highly susceptible to protease degradation compared to subunits p125 and p50. In fact, human pol δ heterotetramer can be converted to the heterotrimeric form through proteosomal degradation of the p12 subunit in response to DNA damage [14]. Degradation of the p12 subunit in response to DNA damage has been reported to be a potential regulatory mechanism utilized by cells that enhances the ability of pol δ to discriminate against damaged bases and mismatched primers [15]. Several reports have shown that polymerase δ can be post-translationally modified by phosphorylation, acetylation, sumoylation, and ubiquitination [16,17]. These modifications have been shown to have profound effects on the function of the polymerase. For instance, the phosphorylation of subunit p66 has been shown to regulate its interaction with PCNA [15]. 

The current study is focused on investigating the functional significance of lysine acetylation on pol δ. Global analyses of protein modifications have revealed that some of the proteins that play a role during Okazaki fragment maturation are acetylated [17]. These proteins can also be acetylated in vitro [18]. In addition to acetylating histones, the acetyltransferase p300 has been shown to also acetylate non-histone proteins altering their functionality. The acetylation of PCNA by p300 has been shown to increase its binding affinity for DNA pol δ [19]. Previous reports have shown that the acetylation of human FEN1 by p300 significantly reduces both its DNA binding efficiency and its nuclease activity [20]. Acetylation has also been demonstrated to increase the helicase and the nuclease activity of human Dna2 [18]. These observations suggest a possible role of acetylation in the promotion of higher-fidelity DNA replication. Presumably, reducing the activity of FEN1 delays ligation until all of the DNA synthesized by the low-fidelity pol α has been removed. The increased nuclease activity of Dna2 keeps the displaced flap short, which in turn facilitates the strand displacement synthesis. In addition, because Dna2 does not cleave down to a nick, the ligation would still be delayed. The notion that the cell utilizes protein acetylation to enhance the fidelity of DNA replication is quite intriguing. 

Herein, we report the first detailed analysis of the effect of acetylation on the function of human DNA pol δ. Our mass-spectrometric analysis has revealed that human pol δ is acetylated on subunits p125 and p12. The current data show that the acetylated form of human pol δ displays a higher polymerization activity compared to the unmodified form of the enzyme. Additionally, the acetylation enhances the ability of the polymerase to resolve complex structures such as G-quadruplexes and other secondary structures that might be present on the template strand. More importantly, the ability of pol δ to displace a downstream DNA fragment is enhanced upon acetylation. Taken together, the current study shows that acetylation significantly impacts the activity of pol δ and supports the hypothesis that acetylation promotes higher-fidelity DNA replication. 

## 2. Materials and Methods

### 2.1. Materials

Oligonucleotides utilized for experimentation were synthesized by Integrated DNA Technologies (IDT, Coralville, IA, USA) and Midland Certified Reagents (Midland, TX, USA). The DNA substrates were 5′ radiolabeled using [γ-^32^P]dATP or 3′ radiolabeled using [α-^32^P]dCTP (6000 Ci/mmol), purchased from PerkinElmer Life Sciences, Waltham, MA, USA. The *Escherichia coli* (*E. coli*) Klenow fragment of DNA polymerase I (for 3′ DNA radiolabeling), the T4 polynucleotide kinase (for 5′ DNA radiolabeling), and ATP were purchased from Roche Applied Science, Indianapolis, IN, USA. All other necessary reagents were purchased from the best available commercial sources.

### 2.2. Substrate Design

Substrates utilized for experimentation were created to reflect the Okazaki fragment intermediates of DNA replication. These DNA substrates are listed in Table 1, with upstream and downstream primer sequences indicated in a 5′-3′ direction and template sequences in a 3′-5′ direction for visual alignment. Primers were 5′ end labeled with [γ-^32^P] ATP using T4 polynucleotide kinase [21]. The primers were then passed through Mini Quick Spin Oligo Columns (Bio-Rad Laboratories, Herakles, CA, USA) to remove the unincorporated radioactive material. The radiolabeled primers were then separated by electrophoresis on a 12% denaturing gel, from which they were then isolated and purified. The substrates used for the synthesis assays were constructed by annealing a labeled upstream primer to the template in a 1:4 ratio in Nuclease-Free Duplex Buffer (IDT, IA, USA). The substrates used for the strand displacement synthesis were constructed by annealing a labeled upstream primer to a template and a downstream primer in a 1:4:6 ratio, respectively [21]. The G-quadruplex substrates were prepared by annealing the respective radiolabeled primer to the G-quad template (Table 1). The annealed substrates were then purified by native page (8%), excised from the gel, and purified. The purified substrate was re-suspended in TE buffer. The substrates were then supplemented with 100 mM KCl and incubated at room temperature for 12 h to allow the formation of an intramolecular G-quadruplex structure. To promote the formation of an intermolecular G-quadruplex structure, the G-quadruplex substrate was supplemented with the G-quad template (1 µM) and incubated at room temperature for 1 h [22]. All the substrates were prepared by combining the primers with their respective templates in annealing buffer, heating the mixture at 95 °C, and then allowing the reactions to cool slowly to room temperature. The substrates used for each assay have been drawn above each gel in the figures.

### 2.3. Purification of Recombinant Proteins

Human DNA polymerase δ (h-pol δ) was expressed and purified as previously described with some minor modifications [12]. *Escherichia coli* (*E. coli*) strain BLR (DE3) cells were co-transformed with pLacRARE2 plasmid, pET-hPold1, and pCOLA-hPold234 and selected on LB agar plates containing ampicillin, kanamycin, and chloramphenicol at 37 °C. A single colony was used to inoculate a 100 mL starter culture that was grown overnight at 37 °C. This overnight culture was used to inoculate 2 L of autoinduction media (at final concentrations each liter contained 90 mM KPi, pH 7.0, 1 g/L glucose, 2 g/L lactose, 0.5% glycerol, 12 g peptone, and 24 g yeast extract), supplemented with ampicillin (100 µg/mL final), kanamycin (50 µg/mL final), and chloramphenicol (30 µg/mL), and grown at room temperature for 48 h before harvesting by centrifugation. The cell pellet was resuspended in 100 mL lysis buffer (40 mM HEPES pH 7.5, 100 mM NaCl, 10% glycerol, 30 mM imidazole, and 0.5 mM PMSF) and lysed by sonication. The cleared lysate was loaded onto a Ni-NTA column, which was equilibrated with Ni-Wash buffer (40 mM HEPES pH 7.5, 100 mM NaCl, and 10% glycerol). Following the washing of the column with 10 column volumes of Ni-Wash buffer, h-pol δ was eluted using a 100 mL linear gradient of 30–300 mM imidazole in Ni-Elution buffer (40 mM HEPES pH 7.5, 100 mM NaCl, 10% glycerol, and 300 mM imidazole). The purity of the fractions containing h-pol δ was analyzed through SDS-PAGE, and these fractions were combined and subjected to overnight dialysis against 2 L of h-pol δ storage buffer (40 mM HEPES pH 7.5, 100 mM NaCl, and 10% glycerol). The dialyzed protein was concentrated and introduced to an SEC column (Bio-Rad, CA, USA) at a flow rate of 0.5 mL/min, which was pre-equilibrated with the h-pol δ storage buffer. All subunits of h-pol δ co-eluted in a single peak, and the fractions containing the protein were pooled, tested for enzyme activity, divided into aliquots, and frozen at −80 °C.

To express recombinant human RFC, pET-hRFC1 and pCOLA-hRFC2345 were co-transformed into BLR (DE3) cells. The co-transformants were selected on LB agar plates containing ampicillin, kanamycin, and chloramphenicol at 37 °C. A single colony was used to inoculate a starter culture of 100 mL, which was grown overnight at 37 °C. The overnight culture was then used to inoculate 2 L of autoinduction media supplemented with ampicillin (100 µg/mL final) and kanamycin (50 µg/mL final) and grown at room temperature for 48 h. The cells were harvested by centrifugation and human RFC was purified as previously described [12].

cDNAs encoding different subunits of RPA were cloned into a pET vector and over-expressed in *E. coli* strain BL21 (DE3) and purified using an Affi-gel blue column as previously described [23]. The mammalian PCNA was expressed and purified as previously described [24,25].

### 2.4. Construction, Expression, and Purification of the Catalytic Domain of h-p300

For the construction of the minimal HAT domain of human p300, the DNA fragment encoding residues 1284–1669 [26] was synthesized by Genscript, Piscatwe, NJ, USA and inserted into the ECoRI-Xho1 site of pUC57-Kan to yield the p300-HAT-pUC57-Kan plasmid. The p300-HAT-pUC57-Kan plasmid was then digested and ligated into the EcoRI-Xho1 site of pET28b to yield the p300-HAT-pET28b plasmid for protein expression. Expression and purification of the HAT domain of human p300 was carried out as previously described with minor modifications [27]. Briefly, competent *E. coli* BL21(DE3) cells were transformed with the p300-HAT-pET28b plasmid. Successful transformants were selected at 37 °C on LB agar plates containing kanamycin. A single colony was used to inoculate 100 mL LB-Kanamycin starter culture, which was grown overnight at 37 °C. The overnight culture was used to inoculate 2 L of autoinduction media (at the final concentrations each liter contained 90 mM KPi, pH 7.0, 1 g/L glucose, 2 g/L lactose, 0.5% glycerol, 12 g peptone, and 24 g yeast extract). The autoinduction media was supplemented with kanamycin (50 µg/mL final). The cells were grown at room temperature for 48 h after which they were harvested by centrifugation. 

The cell pellet was re-suspended in 100 mL lysis buffer (25 mM HEPES pH 7.5, 300 mM NaCl, 10% glycerol, 1 mM β-mercaptoethanol, 30 mM Imidazole, and 0.5 mM PMSF. The re-suspended cells were then lysed by sonication. The cell lysate was then centrifuged to remove the cellular debris and the cleared lysate was loaded onto a Ni-NTA column, which was pre-equilibrated with Ni-Wash buffer (25 mM HEPES pH 7.5, 300 mM NaCl, 10% glycerol, and 10 mM imidazole). After washing the column with 10 cv of Ni-Wash buffer, p300 HAT domain was eluted with 100 mL of a linear gradient of 10–300 mM imidazole in Ni-Elution buffer (25 mM HEPES pH 7.5, 300 mM NaCl, 10% glycerol and 300 mM imidazole. The purity of the p300 HAT domain-containing fractions was assessed by SDS-PAGE. The fractions containing p300 HAT domain were pooled together and dialyzed overnight against p300 storage buffer (25 mM HEPES pH 7.5, 300 mM NaCl, and 10% glycerol). The dialyzed protein was measured for acetyl transferase activity, aliquoted, and stored at −80 °C.

### 2.5. Acetylation of Recombinant h-Pol δ

The acetylated form of pol δ was prepared by mixing 100 nM h-pol δ with the catalytic domain of h-p300 and acetyl-CoA in a 1:1:10 ratio in 1X acetyltransferase (AT) buffer (50 mM Tris-HCl (pH 8.0), 1 mM dithiothreitol, 10 mM sodium butyrate, 10 mM sodium chloride, 1 mM phenylmethylsulfonyl fluoride, and 10% (*v*/*v*) glycerol). The unmodified form of h-pol δ was prepared by adding 100 nM h-pol δ to the 1X HAT buffer. Unmodified and acetylated reactions were incubated at 37 °C for 30 min to create the acetylated and the unmodified forms of pol δ for the primer extension and strand displacement assays. The polymerase acetylation was confirmed both by autoradiography, as well as Western blot analysis utilizing a pan acetyl lysine antibody (Cell Signaling, Danvers, MA, USA, catalog # 9441).

### 2.6. Pol δ Extension, Strand Displacement, and Structure Resolution Assays

To perform the time-course primer extension assays, we incubated 100 nM of either unmodified pol δ (UM-pol δ) or acetylated pol δ (AC-pol δ) with 5 nM of substrate in a reaction buffer consisting of 30 mM Tris HCl (pH 8.0), 1 mM ATP, 50 μM dNTPs, 12.5 mM NaCl, and 4 mM MgCl2 for 0, 2.5, 5, and 10 min. To perform the concentration-dependent assays, we incubated increasing concentrations of UM-pol δ or AC-pol δ (20, 50, 100, 200 nM) with 5 nM of substrate in reaction buffer for 10 min. We terminated the time-course and concentration-dependent assays by adding 2X termination dye (8 mM EDTA, 0.08% SDS, and Formamide to a 50% final concentration). We then heated the terminated reactions at 95 °C and loaded them onto a pre-warmed denaturing polyacrylamide gel (12% polyacrylamide, 7 M urea) and separated the reaction products by electrophoresis for 1 h and 20 min at 80 watts. After drying the gel for 1 h, we exposed it to a phosphor screen overnight and scanned it using a PhosphorImager (GE Healthcare, Chicago, IL, USA). We analyzed and quantified the data using ImageQuant software. The figures in our experiments are representative images, which we performed at least three times using freshly acetylated polymerase in each replicate.

### 2.7. Exonuclease Assays

The exonuclease activity of pol δ was investigated using a primer that contained nucleotide mismatches at various positions (mismatched primer sequences are given in Table 1). Either 5 nM of mismatched primer substrate or of a substrate with the correct nucleotide sequence was mixed with 100 and 200 nM of either UM-pol δ or AC-pol δ in a reaction buffer containing 30 mM Tris HCl (pH 8.0), 1 mM ATP, 12.5 mM NaCl, and 4 mM MgCl_2_ without dNTPs). Reactions were incubated at 37 °C for 10 min, after which they were terminated by the addition of 2X termination dye. The samples from the reaction were subjected to heating at 95 °C and subsequently loaded onto a denaturing polyacrylamide gel (18% polyacrylamide, 7 M urea) that was pre-warmed. The cleavage products were then electrophoresed for an hour at 80 watts. The gel was dried for an hour and exposed to a phosphor screen before being scanned on a PhosphorImager (GE Healthcare, IL, USA). The data were analyzed and quantified using ImageQuant software (GE Healthcare, IL, USA).

### 2.8. Quantification of Polymerization and Exonuclease Products

Accumulation of products were quantified using the ImageQuant TL 7.0 software (GE Healthcare, IL, USA). The pixel densities were calculated by drawing boxes around each product. The local background subtraction was applied to normalize the pixel densities across the gel image.

% of Synthesis Products = {(b)/(b + c + a)} *× 100; where, “b” is the synthesis products, “c” is the exonuclease products, and “a” is the remaining unsynthesized substrate.

% of Strand Displacement Products = {(d)/(d + c + b + a)} × 100; where, “d” is the strand displacement products, “b” is the synthesis products, “c” is the exonuclease products, and “a” is the remaining unsynthesized substrate. 

% of Structure Resolution Products = {(f)/(f + c + b + a)} × 100; where, “f” is the structure resolution products, “b” is the synthesis products, “c” is the exonuclease products, and “a” is the remaining unsynthesized substrate. 

% of Fully Synthesized Product = {(e)/(e + d + c + b + a)} × 100; where, “e” is the full-length product, “d” is the strand displacement products, “b” is the synthesis products, “c” is the exonuclease products, and “a” is the remaining unsynthesized substrate. 

% of Exonuclease Products = {(c)/(c + a)} × 100; where, “c” is the exonuclease products, and “a” is the remaining uncleaved substrate. 

### 2.9. Mammalian Cell Culture

ATCC-purchased human embryonic kidney (HEK293) cells (CRL-1573) were maintained in Minimum Essential Media (MEM) complemented with 10% fetal bovine serum (FBS), 2 mM L-Glutamine, 1000 units/mL penicillin, and 100 μg/mL streptomycin. The cells were cultured at 37 °C in a 5% CO_2_ humidified environment until they attained 80% confluency. Subsequently, the cells were lysed with RIPA buffer containing 50 mM sodium butyrate, and the protein concentration in the lysate was estimated using the Bradford Assay (Bio-Rad, CA, USA).

### 2.10. IP and Western Blot Analysis of h-pol δ

Immunoprecipitation was conducted using a modified version of the protocol provided in the Dynabead protein G manual from Thermo Fisher Scientific, Waltham, MA. Initially, 20 μL of either control rabbit IgG (Santa Cruz Biotechnology Inc., Dallas, TX, USA, sc-2027) or antibodies to acetyl-lysine (Cell Signaling, MA, USA, #9441) were bound to 1 mg of HEK293 whole-cell extract in 200 μL of 1X PBST with end-over mixing for 1 h at room temperature. Dynabeads (50 μL) were then added to the cell lysates and incubated with end-over mixing for 30 min at room temperature. The Dynabeads–antibody–antigen complex was washed thrice with 200 μL of washing buffer and magnetically separated between washes. For elution, 20 μL of elution buffer and 20 μL of premixed 2X NuPAGE LDS sample buffer with NuPAGE sample reducing agent were added, followed by heating the samples at 70 °C for 10 min. The immunoprecipitate was magnetically separated, and the supernatant was loaded onto precast 7.5% TGX gels (Bio-Rad) for the Western blot analysis. Antibodies used for Western blotting included DNA pol δ cat antibody (Santa Cruz Biotechnology Inc., sc-8797), DNA pol δ 2 antibody (Santa Cruz Biotechnology Inc., sc-390583), PolD3/p66 antibody (Bethyl Laboratories, Inc., Montgomery, MA, USA, catalog # A301-244A), and anti-POLD4/p12 antibody (MyBiosource, San Diego, CA, USA, catalog # MBS2541031).

### 2.11. Analysis of the Acetylation Status of h-pol δ by Mass Spectroscopy

The study utilized an LTQ-Orbitrap Velos mass spectrometer running XCalibur 2.2 SP1 (Thermo Scientific, Waltham, MA, USA) to collect tandem mass spectra from both in vitro acetylated recombinant h-pol δ containing all four subunits and immunoprecipitates of endogenous h-pol δ from HEK293T cells, using a data-dependent method that targeted the top 15 MS/MS signals. The enzyme specificity was set to trypsin, allowing up to two missed cleavages. High-scoring peptide identifications met certain criteria, including a delta CN value of ≥ 0.10, cross-correlation (Xcorr) values of ≥ 1.5, and precursor accuracy measurements within ±3 ppm in at least one injection. Precursor and product ions were measured with mass accuracies of ±10 ppm and 0.8 Da, respectively. The fixed modification of carboxamidomethyl cysteine was specified, while dynamic modifications such as oxidized methionine and acetylation of lysine residues were permitted. To classify acetylated proteins, gene ontology (GO) annotations from Uniprot were used.

## 3. Results

### 3.1. DNA Polymerase δ Is Acetylated In Vitro and In Vivo

The current study was focused on understanding the effects of acetylation on the functional properties of human DNA polymerase delta. Since the acetyltransferase p300 was shown to mediate the acetylation of other Okazaki fragment maturation proteins including FEN1 and Dna2 [18,20], the acetylation of pol δ was also analyzed by incubating the purified recombinant protein in the presence of full-length h-p300 and acetyl-CoA. Western blot analysis with a pan acetyl lysine antibody revealed subunit p125 is likely acetylated (lane 3, Figure 1A). The autoacetylation of the acetyltransferase, p300, was also observed (lane 4, Figure 1A). In vivo acetylation of pol δ was evaluated by treating HEK293 cellular lysate with a pan acetyl lysine antibody. The immunoprecipitates were then separated on a 4–15% SDS-PAGE, and then analyzed using an antibody against p125. Our results suggest that the catalytic subunit of pol δ is endogenously acetylated (lane 2, Figure 1B). Using Western blot, we did not detect acetylation on the other subunits of pol δ either in our in vitro acetylated polymerase or in the endogenous cellular polymerase. We further evaluated the sites of lysine acetylation on both the in vitro acetylated protein, as well as immunoprecipitated endogenous polymerase. Our mass spectrometry analysis using in vitro acetylated protein, indicate that K16, K897, and K998 are modified by acetylation on PolD1 (p125 subunit) and K15 and K25, on PolD4 (p12 subunit) (Appendix A). Of the identified acetylation sites, residues K897 and K998 were previously reported to be also ubiquitinated in vivo [28,29,30]. Additionally, K1007 on the catalytic subunit has been previously reported to be both acetylated and ubiquitinated [30,31,32] (Figure 1B). Similarly, K15 and K25 were both shown to be ubiquitinated [29,30]. Lysine residues K439 on PolD1 and K25 on PolD4 were found to be modified in the endogenous form of the protein (Appendix A). We did not detect lysine acetylation on the other two subunits (p50 and p66) of the polymerase. However, mass spectrometry studies have reported nine lysine acetylation sites on PolD3 (p66 subunit) on the online repository PhosphoSite Plus [33].

### 3.2. Acetylation Stimulates the Synthesis Activity of DNA Polymerase δ

Proliferating cell nuclear antigen A (PCNA) stimulates the processive synthesis of DNA polymerase δ [34]. Since acetylated forms of Okazaki fragment maturation proteins displayed an alteration in their enzymatic activities, we investigated whether the acetylation of pol δ regulates its synthesis activity and alters its processivity in the absence of PCNA. For these studies, a 44 nt upstream primer was radiolabeled on the 5’ end and annealed to 110 nt long template. The structure of the substrate is depicted above Figure 2A. The ability of the unmodified (UM) and the acetylated (AC) forms of the polymerase to extend this primer and form full-length products was then analyzed. The time-course assay revealed that both forms of the enzyme produced a ladder of products whose concentration increased as time progressed (Figure 2A). Time sampling of the reactions showed that both forms of the polymerase pause more at some sequences (GC-rich sequences) than others. However, at all-time points tested, the AC-pol δ was more active than the UM-pol δ (red arrow vs. black arrow). It was indeed striking to observe the enhanced processive synthesis with the acetylated form of the enzyme compared to the unmodified form. This was evidenced by the presence of more full-length extension products in the acetylated form of pol δ compared to the unmodified form (Figure 2B). We also monitored the synthesis activity using varying concentrations of the UM and AC-pol δ (Figure 2C). The presence of more full-length extension products was even more apparent with an increasing protein concentration, in the acetylated form compared to the unmodified form of pol δ. The quantitation of the percent of full-length synthesis revealed a ~3-fold increase in the formation of full-length products in the acetylated form of pol δ compared to the unmodified form. Our results suggest that even in the absence of PCNA, acetylation stimulates the activity of pol δ and enhances its ability to generate more full-length extension products. The incubation of pol δ with either p300 alone or acetyl CoA alone did not alter the enzymatic activity of the polymerase (Appendix A). This indicates that the enhanced polymerase function can be specifically attributed to the lysine acetylation. 

### 3.3. Acetylation Stimulates the Strand Displacement Activity of Pol δ

We next evaluated the effect of acetylation on the ability of pol δ to displace a downstream blocking sequence. For these studies, a 44 nt primer, radio-labelled on the 5′ end, and a 60 nt downstream blocking sequence were annealed to a 110 nt template. The sequence of the downstream blocking sequence was randomly designed with approximately equal AT/GC content. A 6 nt gap was left between the upstream and the downstream sequences as depicted in Figure 3A. The ability of the UM and AC forms of pol δ to extend the upstream primer through the gap, and into the downstream sequence, displacing this sequence, was analyzed. Our results show that both forms of the polymerase are able to extend the upstream primer to fill in the 6 nt gap, and then there is considerable stalling that is observed when the 5’ end of the downstream blocking sequence is encountered (Figure 3A). A further extension of the primer necessitates the displacement of the downstream blocking sequence, generating higher molecular weight extension products. Even though full-length synthesis was not achieved in this particular assay, it was apparent that acetylation markedly stimulated the strand displacement activity of pol δ, and this was evidenced by the generation of higher molecular weight products in the acetylated form of pol δ, compared to the unmodified form. The quantitation of the relative strand displacement activity revealed a ~2-fold increase in the amount of higher molecular weight products in the acetylated form of pol δ compared to the unmodified form.

Since the substrate sequence played a role in the ability of pol δ to perform strand displacement synthesis, we were curious as to whether the polymerase could fully extend the upstream primer and displace the entire downstream blocking sequence. For these studies, we modified the downstream blocking sequence in such a way that its AT content was considerably higher, as depicted in Figure 3B. We then examined the ability of the modified and the unmodified forms of the polymerase to extend the upstream primer through the 6 nt gap and into the downstream blocking sequence, displacing it. As we had anticipated, both forms of the polymerase were able to displace the entire downstream blocking sequence and generate full-length primer extension products. Interestingly, we observed a ~4-fold increase in the amount of full-length products generated by the AC-pol δ compared to the UM-pol δ (lanes 5 and 9, Figure 3B). We also observed considerable stalling with the UM-pol δ compared to the AC-pol δ (lanes 5 and 9, Figure 3B). When the unmodified form of pol δ encounters the 5′ end of the downstream blocking sequence, its ability to displace this sequence seems to be repressed, more so in the absence of its accessory proteins. However, with acetylation, the polymerase is rapidly able to displace the downstream sequence and generate more full-length extension products.

Previous reports have shown that the presence of GC-rich regions on the downstream blocking sequence greatly impede the strand displacement activity of pol δ [35]. We evaluated the ability of the modified and the unmodified forms of the polymerase to displace a downstream blocking sequence that contained ~80% GC content. The design of this substrate is depicted in Figure 3C. It was not surprising that the ability of both forms of the enzyme to displace this downstream sequence was greatly inhibited, as full-length extension products were not generated (Figure 3C). However, with acetylation, higher molecular weight products were observed compared to the unmodified form of pol δ. Relative strand displacement analysis revealed ~1.8-fold high molecular weight products with the acetylated form of pol δ compared to the unmodified form. Our current results provide convincing evidence that acetylation of pol δ markedly stimulates its ability to displace downstream blocking sequences.

### 3.4. Acetylation Enhances the Ability of Pol δ to Resolve Complex Secondary Structures

Secondary structures appearing on the template strand present a great challenge to the replication process [9]. Since our results indicated that acetylation stimulates the strand displacement activity of pol δ, we envisioned that acetylation would also stimulate the structure resolution activity of the polymerase. For these assays, the template was designed in such a way that it contained 18 nt complementary region, which would form a short stem, and a 10 nt non-complementary region, which would form a loop. On the 3’ end of the loop structure was a 14 nt tail. The 44 nt primer, which was radio labeled on the 5’ end, was annealed to this template leaving 6 nt gap between the primer terminus and the loop structure. The design of the substrate is depicted in Figure 4A. Our results show that the unmodified and the modified forms of pol δ rapidly synthesize the 6 nt gap region that is upstream of the loop structure. Upon encountering the secondary structure, there is significant stalling observed on both forms of the polymerase (Figure 4A). This is evidenced by the striking accumulation around the 50 nt product. The synthesis process then proceeds through the loop structure, resolving the structure, and producing the ladder of extension products. The absence of primer extension intermediates past the loop region (above 68 nt) indicates a rapid extension process as the enzymes are no longer facing the challenge posed by the presence of the loop structure. Our results indicate that the acetylation of pol δ stimulates its ability to resolve secondary structures on the template strand, which leads to the formation of more full-length extension products compared to the unmodified form. The relative strand displacement activity of the polymerase was ~2 fold higher with the acetylated form of the enzyme compared to the unmodified form. 

G-rich sequences that are present on a DNA strand have the propensity to associate and form G-quartet structures, which are stabilized by the presence of monovalent cations such as K^+^ or Na^+^. The stacking of two or more G-quartet structures forms a G-quadruplex structure. The presence of these complex structures on a DNA strand poses significant challenges to the entire replication machinery [22]. We evaluated the ability of the unmodified and the modified forms of pol δ to resolve these structures and generate full-length primer extension products. For these studies, we utilized two substrates. One substrate contained a G-quartet structure (depicted in Figure 4B), and the second substrate contained a G-quadruplex structure (depicted in Figure 4C). With the G-quartet substrate, both forms of the polymerase were able to extend the 44 nt primer that was annealed upstream to the structure. However, we observed a significantly higher content of full-length extension products with the acetylated form of pol δ compared to the unmodified form (Figure 4B). We observed considerable stalling of the polymerase, both modified and unmodified form, upon encountering the G-quartet structure (denoted by the gray arrow, Figure 4B). Our results show an enhanced ability of the acetylated form of pol δ to resolve the secondary structure and generate a higher content of full-length products. 

Compared to the G-quartet, the presence of a G-quadruplex structure on the template strand posed a significant challenge to the modified and the unmodified forms of pol δ (Figure 4C). The primer extension ability of the enzymes was substantially inhibited, and the inhibition was much more apparent in the unmodified form of the enzyme compared to the acetylated form. It was quite interesting to observe that with acetylation, the polymerase can resolve the complex G-quadruplex structure and generate full-length extension products (110 nt product). 

### 3.5. Replication Accessory Proteins Can Stimulate Acetylated Polymerase

During lagging strand synthesis, pol δ is aided by PCNA and RFC to perform processive synthesis [36,37]. Thus, in the presence of these two accessory proteins, the synthesis activity of pol δ is greatly enhanced. Similarly, RPA can assist pol δ while synthesizing through barriers on the genome or through structured regions [38,39]. RFC loads PCNA onto the DNA in an ATP-dependent manner. Nonetheless, in an in vitro assay utilizing an unblocked linear substrate, PCNA can slide onto the substrate and engage with pol δ. Since AC-pol δ displayed higher synthesis, strand displacement, and structure resolution, we were interested to test the ability of the individual accessory proteins to further enhance the function of pol δ. We used a limiting concentration of pol δ in our assays compared to the accessory proteins to test the threshold for structure resolution. Both forms of the pol δ (UM and AC) showed very little synthesis at a 10 nM concentration (lane 2 and lane 10, Figure 5). However, in the presence of the accessory proteins, the UM-pol δ displayed a ~4.5-fold stimulation in structure resolution in the presence of PCNA (compare lanes 3–4 to lane 2), a ~2-fold stimulation in the presence of RFC (compare lanes 5–6 to lane 2), and a ~3.8-fold stimulation in the presence of RPA (compare lanes 7–8 to lane 2). We observed similar levels of stimulation in the activity of AC-pol δ in the presence of accessory proteins (compare lanes 11–16 to lane 2). With the acetylated form of the polymerase, we also observed a significantly higher amount of full-length synthesis (110 nt product). 

### 3.6. Acetylated Polymerase Displays Higher Exonuclease Activity

The synthesis and exonuclease activity are competing functions for the polymerase. Deletion of the exonuclease domain allows the polymerase to perform more processive synthesis [12,40,41,42]. In our previous assays (Figure 2, Figure 3 and Figure 4), we noticed a higher percentage of exonuclease activity in the presence of the acetylated form of the polymerase. Since polymerization and exonuclease activities are finely balanced during replication, we were interested in testing how the acetylation influences the nuclease activity of the polymerase. We first assayed for nuclease activity in the absence of polymerase, but in the presence of p300. We did not observe any nuclease activity when p300 was incubated alone with the substrate, suggesting that the increased exonuclease activity we observed in our AC-pol δ lanes was not resultant of a contaminating nuclease from the acetyltransferase. We created substrates containing mismatches either 3, 5, or 7 nucleotides away from the 3′ end of the primer terminus and tested exonuclease function. For each substrate, we observed a higher exonuclease activity in the presence of the AC-pol δ compared to the UM-pol δ (Figure 6A). Since polymerization and exonuclease are competing activities, we tested the synthesis activity in the early phase of the polymerization reaction utilizing a higher concentration of the dNTPs (150 µM). While both the UM and AC-pol δ displayed similar levels of exonuclease activity (Figure 6B), we observed a significant stimulation of synthesis in the early phase of the reaction by the AC-pol δ. This suggests that when dNTPs are limiting, the AC-pol δ displays increased exonuclease activity; however, when normal levels of dNTPs are used in the reaction, its preferred function is synthesis activity. 

## 4. Discussion

Previous reports have shown that the acetylation of some proteins involved in the Okazaki fragment maturation process alters the path through which the fragments are processed [18]. The acetylation of FEN1 greatly inhibits its catalytic activity, while acetylation of DNA2 stimulates its catalytic activity [18,20]. The consequence of this mode of regulation is the formation of longer flaps, which are then efficiently processed by the enhanced nuclease activity of acetylated DNA2 [18]. The formation of longer flaps during the Okazaki fragment maturation process, ensures the complete removal of the initiator primer that is synthesized by the error prone DNA polymerase α [9]. There is accumulating evidence that supports the notion that acetylation promotes high-fidelity DNA replication, which consequently improves genome stability [18,20,43]. The results presented in this study are highly in favor of this idea. Our current results have established that acetylation positively regulates the functions of human DNA pol δ. 

We first evaluated the acetylation status of human DNA pol δ in vivo and in vitro. The heterotetrameric enzyme harbors its catalytic activity on the large subunit, p125 [44]. The enrichment of acetylated proteins with a pan acetyl lysine antibody, followed by probing for p125 by standard Western blot procedures, provides evidence that this subunit is acetylated in vivo. This observation prompted us to further investigate the relevance of this modification on the catalytic activity of pol δ. A thorough analysis of the lysine acetylation sites using in vitro acetylated polymerase, by mass spectrometry, revealed four sites that are modified on the catalytic subunit. These sites include K16, K439, K897, and K998. The strict conservation of these lysine residues among higher eukaryotic organisms implicate a crucial role played by these residues in polymerase activities. Structural evaluation of the catalytic subunit of yeast pol δ in complex with a template primer and an incoming nucleotide shows that the N-terminal region of the polymerase interacts with the last visible nucleotide of the unpaired segment of the template strand [45]. The presence of a positively charged lysine residue creates favorable electrostatic interactions with the negatively charged DNA. The loss of the positive charge by acetylation provides a convincing argument for the increased synthesis activity that is observed in acetylated pol δ. Arguably, the modulation of the polymerase-DNA interaction near the N-terminal region, by acetylation, allows for faster and more processive polymerization activity. 

DNA polymerase δ plays an indispensable role in the Okazaki fragment maturation process [46]. As the polymerase extends the initiator RNA/DNA primer, it encounters the downstream Okazaki fragment and displaces it into a 5′ flap structure [47]. The processing of the flap is either facilitated by the unilateral action of FEN1 via the short flap pathway, or by the combined action of DNA2 and FEN1 via the long flap pathway [46]. We designed substrates that mimic Okazaki fragments and investigated how modification of pol δ by acetylation might influence its ability to displace a downstream blocking sequence. As would be expected, the nature of the downstream blocking sequence affects the strand displacement activity of pol δ. For instance, increasing the GC content of the downstream sequence significantly inhibits the strand displacement activity of the polymerase. A complete displacement of the downstream sequence could not be achieved with this substrate. The presence of high molecular weight synthesis products beyond the 6 nt gap indicates the extent to which the downstream sequence has been displaced. Interestingly, our results show that acetylation enhances the ability of polymerase δ to displace a larger portion of the downstream sequence. On the contrary, AT-rich sequences are fairly easy to displace, and would therefore not pose a significant barrier to the strand displacement activity of pol δ. Indeed, when we increased the AT content of the downstream sequence, the polymerase was able to displace the entire sequence and generate full-length products of synthesis. The acetylated form of pol δ was much more effective at generating full-length products compared to the unmodified form. Thus, our current findings highlight a crucial role played by acetylation in enhancing the strand displacement activity of pol δ. In the context of Okazaki fragment maturation process, an enhanced strand displacement activity is highly desirable as it contributes immensely to genome stability. The initiating RNA/DNA primers in Okazaki fragments are laid down by the low-fidelity DNA polymerase α, which lacks any proofreading ability [48]. Errors generated by polymerase α are proofread by polymerase δ, thereby maintaining genome stability [49]. With the current findings, the stimulation of the strand displacement activity by acetylation ensures that the pol α synthesized portion of the lagging strand is completely removed and re-synthesized by the high-fidelity pol δ. Our proposal correlates with the previous observations that acetylation alters the flap-processing activities of FEN1 and DNA2 by promoting the formation of longer flaps, which are stabilized by RPA.

Regulation of a protein’s function by acetylation has been reported in a number of proteins involved in DNA replication and repair, and the effects of this modification are unique to the respective protein that is modified. Acetylation influences cellular processes in a variety of mechanisms, such as altering the DNA-binding properties of some proteins, regulating protein–protein interactions and catalytic properties of some proteins, and it also influences protein stability [50,51]. The nucleotide excision repair factor, XPA has been shown to be regulated by acetylation at lysines 63 and 67, and this modification has a negative effect on the DNA-binding properties of XPA [51]. The acetylation of p53 has been reported to trigger a conformational change that stimulates its DNA-binding ability [52]. The transcription factor E2F1 is also regulated by acetylation [53]. It has been shown that acetylation of E2F1 increases its DNA-binding ability as well as its ability to activate transcription [53]. Acetylation is a regulatory mechanism for the processivity factor PCNA, which impacts PCNA’s ability to slide on DNA in the presence of DNA damage. This modification promotes homologous recombination, which is linked to sister chromatid cohesion [34].

In our current analysis, we have established that acetylation allows the interaction between the polymerase and DNA to be much more effective. It promotes the melting of the downstream duplex DNA so that polymerization can proceed beyond the gap. As discussed above, the N-terminal region of pol δ interacts with the template strand a few nucleotides beyond the primer–template terminus. The lysine acetylation on the N-terminal region would modulate the interaction between the polymerase and DNA in a way that promotes replication. Consistent with this idea, our results show that acetylation stimulates the ability of the polymerase to resolve secondary structures that are present on the template strand. During Okazaki fragment synthesis, the single-stranded region between the upstream and the downstream primers can fold back into a stem-loop structure that would need to be resolved first, before the polymerase can continue extending the upstream primer. The formation of secondary structures on the DNA can have adverse effects on the replication process. These structures can cause replication stalling, which can consequently lead to genome instability [54,55]. A stem-loop region present on the template strand can be viewed as a double-stranded region that would require the strand displacement activity of the polymerase. It is indeed intriguing that acetylation enhances the ability of the polymerase to melt the secondary structure and synthesize full-length products. These findings further support our working hypothesis that the acetylation of polymerase δ promotes genome stability. 

Stretches of DNA or RNA that are rich in guanine nucleotides are prone to forming stable secondary structures known as G-quadruplexes [56]. These structures are widespread throughout the human genome, especially at the telomeric regions, which contain TTAGGG/AATCCC repeats [22,56]. Formation of G-quadruplex structures at the promoter regions have been shown to be involved in the regulation of transcription [57,58,59]. The presence of these structures present significant challenges to replicating polymerases [22]. In fact, in vitro biochemical analysis demonstrated that G-quadruplex structures on the template strand strongly inhibit the synthesis activity of DNA polymerase δ [60]. Replication fork stalling caused by G-quadruplex structures within the telomeric region may lead to telomere instability, which can consequently decrease the overall stability of the human genome [22,61]. As such, it is critical that the cell would evolve a mechanism that would improve the replication process through G-rich sequences. The current studies suggest acetylation to be one such regulatory mechanism that could alter the enzymatic activity of the core participants of DNA replication and repair pathways. Our studies demonstrate that the acetylation of DNA polymerase δ stimulates its ability to resolve complex G-quadruplex structures and synthesize the DNA strand all the way to the end of the template. The ability to resolve these structures is of great value, especially in the context of promoting genome stability. In essence, our in vitro results suggest that the acetylation of pol δ may facilitate the recovery of stalled replication forks caused by G-quadruplex structures, and thus plays an important role in the maintenance of genome stability. 

The current studies have provided the first analysis of the effect of acetylation on the functional properties of pol δ. Further detailed studies are underway, which will evaluate the exact mechanism by which acetylation of pol δ influences its catalytic properties. These studies involve mutational analysis of the lysine sites that are acetylated. Structural analysis of acetylated pol δ in complex with DNA will provide significant insights into the mode of interaction of the acetylated polymerase and DNA substrate.

## Figures and Tables

**Figure 1 genes-14-00774-f001:**
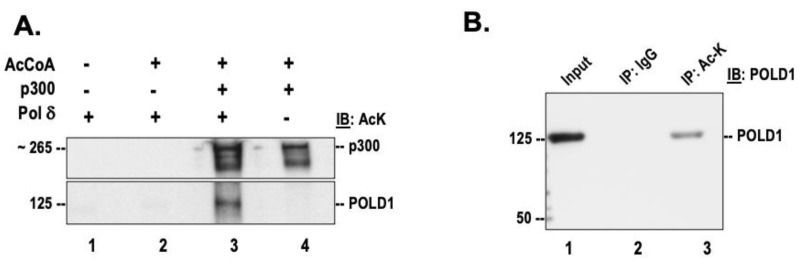
DNA polymerase δ is acetylated in vitro and in vivo. (**A**) Purified recombinant pol δ was incubated in the presence of h-p300 and acetyl CoA and then separated on 4–15% SDS-PAGE, as described in the “Section 2.1”. The acetylated subunits of pol δ (lane 3) were then detected by Western blotting using a pan acetyl lysine (AcK) antibody. (**B**) Whole-cell lysates prepared from HEK293T cells were immunoprecipitated with IgG or AcK antibody. The immunoprecipitated proteins were then separated on 4–15% SDS-PAGE and analyzed by Western blotting using anti POLD1 (p125); 10% of input from the whole-cell lysate (lane 1) served as the control.

**Figure 2 genes-14-00774-f002:**
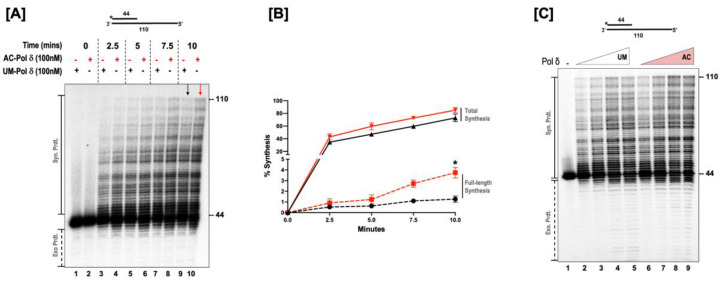
Acetylation enhances the synthesis activity of pol δ. (**A**) The synthesis activity of pol δ was assayed using the 5 nM substrate (U_1_:T_1_) in the presence of 100 nM of either the unmodified (UM) or acetylated (AC) forms of pol δ. Time-course synthesis assay was monitored at the respective time points (0, 2.5, 5, 7.5, and 10 min) as indicated in the figure and as described in the “Section 2.6”. The substrate is depicted above the gel image, with the asterisk denoting the location of the radiolabel. + indicates the presence and—indicates the absence of the respective forms of pol δ. (**B**) Graphical plot of the percent of synthesis against time in minutes. Dashed red lines represent full-length synthesis by AC-polδ and dashed black lines represent full-length synthesis by UM-pol δ. Solid red lines represent total synthesis by AC-polδ and solid black lines represent total synthesis by UM-pol δ. The *p* value (generated by GraphPad Prism) was calculated by an unpaired, two-tailed *t*-test, * = *p* < 0.05. (**C**) Varying concentrations of either acetylated or unmodified pol δ (20, 50, 100, and 200 nM) were assayed for the synthesis activity of both forms of the polymerase. The reactions were set up as described in the “Section 2.6”. The substrate is depicted above each gel image and the asterisk denotes the location of the radiolabel. Exonuclease products (Exo. Prdts.) and synthesis products (Syn. Prdts.) are indicated adjacent to the gel image. The nucleotide sizes are also indicated.

**Figure 3 genes-14-00774-f003:**
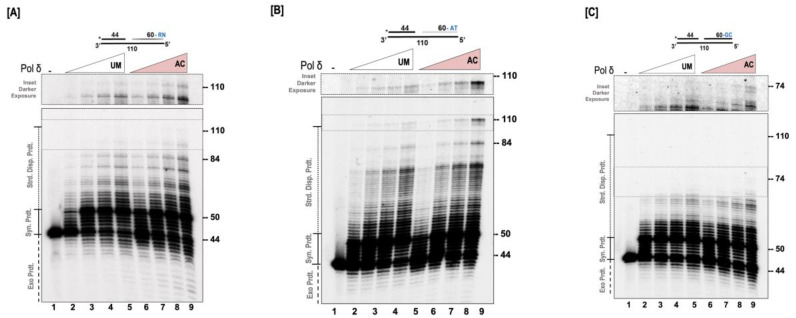
Acetylation stimulates the strand displacement activity of pol δ. The strand displacement activity of pol δ was assayed using the 5 nM substrate at varying concentrations of either unmodified or acetylated pol δ (20, 50, 100, and 200 nM) as described in the “Section 2.6”. (**A**) Substrate (U_1_:D_1_: T_1_) contains a randomly (RN) generated downstream blocking sequence. (**B**) Substrate (U_1_:D_3_: T_3)_ contains an AT-rich downstream blocking (AT) sequence. (**C**) Substrate (U_1_:D_2_: T_2)_ contains a GC-rich downstream blocking (GC) sequence. The substrate is depicted above each gel image and the asterisk denotes the location of the radiolabel. Exonuclease products (Exo. Prdts.), synthesis products (Syn. Prdts.), and strand displacement products (Strd. Disp. Prdts.) are indicated adjacent to the gel image. Nucleotide sizes are also indicated.

**Figure 4 genes-14-00774-f004:**
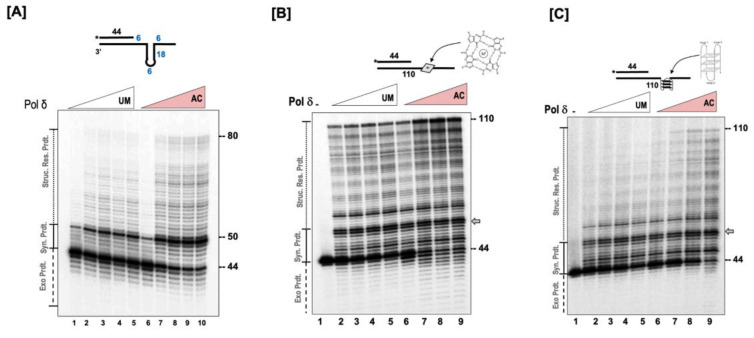
Acetylation stimulates the structure resolution activity of pol δ. (**A**) The structure resolution activity of pol δ was assayed using 5 nM of substrate containing a fold-back in the template (U_1_:T_4_) incubated with the either unmodified (UM) or acetylated (AC) forms of pol δ (20, 50, 100, 200, 400 nM) as indicated in the figure and as described in the “Section 2.6”. The concentration-dependent structure resolution activity of pol δ was assayed using (**B**) a substrate (U_1_:T_5_) containing a G-quartet in the template or (**C**) a substrate (U_1_:T_6_) containing a G-quadruplex in the template at varying concentrations of either unmodified or acetylated pol δ (20, 50, 100, and 200 nM) as described in the “Section 2.6”. The substrate is depicted above each gel image and the asterisk denotes the location of the radiolabel. Exonuclease products (Exo. Prdts.), synthesis products (Syn. Prdts.), and strand displacement product (Strd. Disp. Prdts.) are indicated adjacent to the gel image. Nucleotide sizes are also indicated.

**Figure 5 genes-14-00774-f005:**
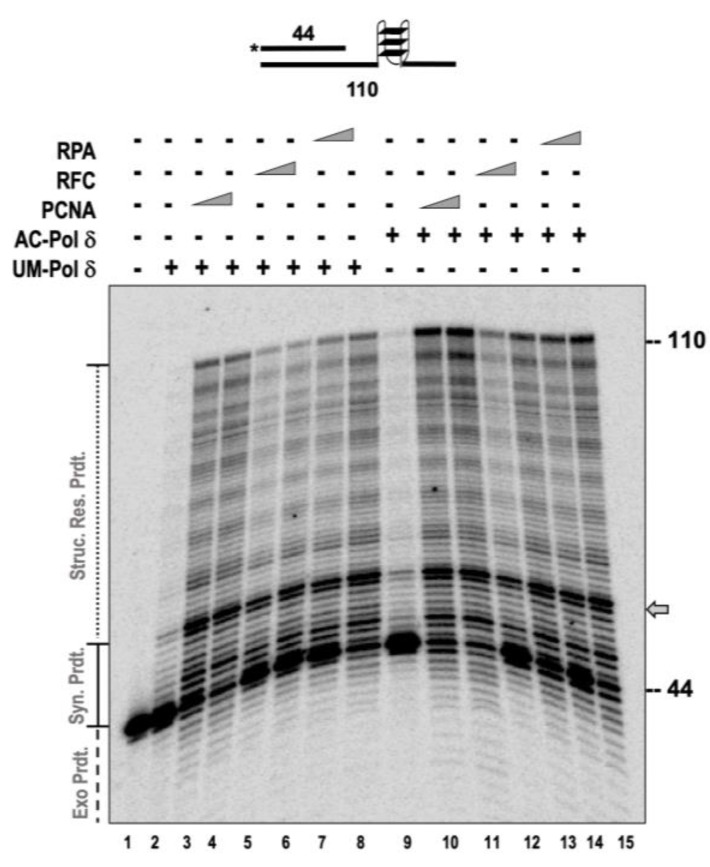
Accessory proteins stimulate the structure resolution activity of UM and AC-pol δ. The structure resolution activity of pol δ was assayed on 5 nM substrate (U_1_:T_6_) containing a G-quadraplex in the template in the presence of 10 nM of UM or AC-pol δ. Increasing the concentrations of PCNA (200 and 400 nM), RFC (100 and 300 nM), or RPA (100 and 200 nM) were incubated along with the substrate and both forms of the polymerase. The substrate is depicted above the gel image and the asterisk denotes the location of the radiolabel. Exonuclease products (Exo. Prdts.) and structure resolution product (Strd. Res. Prdts.) are indicated adjacent to the gel image. Nucleotide sizes are also indicated. Gray arrows denotes position of stalling.

**Figure 6 genes-14-00774-f006:**
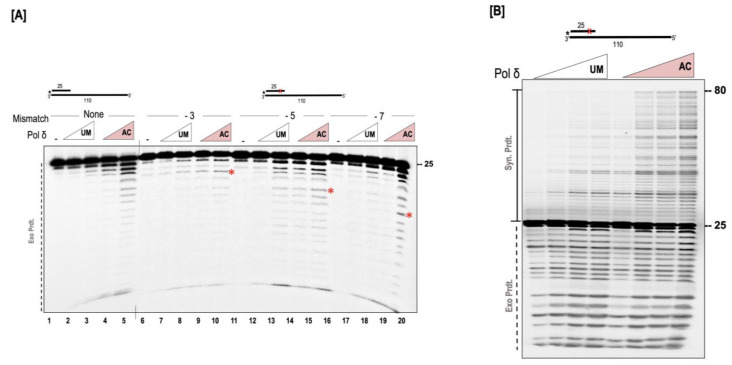
AC-pol δ displays higher exonuclease activity. (**A**) The exonuclease activity of polymerase (100 and 200 nM) was assayed on a 5 nM substrate containing no mismatch (U_2_:T_7_), or containing mismatch 3 nt (U_3_:T_7_), 5 nt (U_4_:T_7_), or 7 nt (U_5_:T_7_) away from the primer terminus as described in the “Section 2.7”. (**B**) The synthesis and exonuclease activity was tested on a 5 nM substrate containing a mismatch 5 nt away from the primer terminus (U_1_:T_6_) in the presence of 10, 50, 100, 200 nM UM or AC-pol δ. The reaction was incubated for 2 mins similar to that described in the “Section 2.7”, with the exception that the polymerase buffer contained 150 µM of dNTPs. The substrate is depicted above the gel image and the asterisk denotes the location of the radiolabel. Exonuclease products (Exo. Prdts.), and synthesis products (Syn. Prdts.) are indicated adjacent to the gel image. Nucleotide sizes are also indicated. Red cross denote position of mismatch.

**Table 1 genes-14-00774-t001:** Oligonucleotide sequences.

Primer	Length	Sequence
Upstream (5′-3′)		
U1	44	GTCCACCCGACGCCACCTCCTGCCTTCAATGTGCTGGGATCCTA
U2	25	GTCCACCCGACGCCACCTCCTGCCT
U3	25	GTCCACCCGACGCCACCTCCTGACT
U4	25	GTCCACCCGACGCCACCTCCCGCCT
U5	25	GTCCACCCGACGCCACCTACTGCCT
Downstream (5′-3′)		
D1	60	AGACGAATTCCGGATACGACGGCCAGTGCCGACCGTGCCAGCCTAAATTTCAATCCACCC
D2	60	GGGCGACTCCCGGGGGCGCCGGCCCGTGCCGGCCGTGCCGGCCTCCCTGTCAACCCACCC
D3	60	AAACTAATTCTGAATAAGAAAAAAAGTGTTTAACTTAAAAGCCTAAATTTAAATCCAAAA
Template (3′-5′)		
T1	110	CAGGTGGGCTGCGGTGGAGGACGGAAGTTACACGACCCTAGGATGTTGGTTCTGCTTAAGGCCTATGCTGCCGGTCACGGCTGGCACGGTCGGATTTAAAGTTAGGTGGG
T2	110	CAGGTGGGCTGCGGTGGAGGACGGAAGTTACACGACCCTAGGATGTTGGTCCCGCTGAGGGCCCCCGCGGCCGGGCACGGCCGGCACGGCCGGAGGGACAGTTGGGTGGG
T3	110	CAGGTGGGCTGCGGTGGAGGACGGAAGTTACACGACCCTAGGATGTTGGTTTTGATTAAGACTTATTCTTTTTTTCACAAATTGAATTTTCGGATTTAAATTTAGGTTTT
T4	80	CAGGTGGGCTGCGGTGGAGGACGGAAGTTACACGACCCTAGGATGTGGTTCTGCCGGACGAGTATTATCTCGTCCGGTAAAGT
T5	110	CAGGTGGGCTGCGGTGGAGGACGGAAGTTACACGACCCTAGGATGTTGGTTAGGGTTAGGGTTAGGGTTAGGGGTCACGGCTGGCACGGTCGGATTTAAAGTTAGGTGGG
T6	110	CAGGTGGGCTGCGGTGGAGGACGGAAGTTACACGACCCTAGGATGTTGGTTCTGTCCAGAGCTCCGGACGAGTATTATCTCGTCCGGAGCTCTGGATAAAGTTAGGTGGG
T7	80	CAGGTGGGCTGCGGTGGAGGACGGAAGTTACACGACCCTAGGATGTTGGTTCTGCTTAAGGCCTATGCTGCCGGTCACGG

Underlined nucleotides represent mismatched nucleotides.

## Data Availability

All original data and images will be provided upon request by the corresponding author.

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
