# Peer review of "DNA Polymerase Delta Exhibits Altered Catalytic Properties on Lysine Acetylation"

_genes, 2023, doi:10.3390/genes14040774_

Round 1
Reviewer 1 Report
This study about the effect of acetylation on the activity of Polymerase delta reveals a lot of information on how Pol delta functions. Most of the data has been represented properly but there are few issues which should be addressed.
The Introduction states subunits p125, p68 and p12 are acetylated. But Results state only p125 and p12 are acetylated. Please check.
Fig 3 & 4: The quantification method is not clear. How were the UM and AC-Pol delta activity compared from the gels? Total product synthesized or the largest product synthesized? Please provide a graphical representation of the quantification.
Fig 4A: For the stem-loop assay, it seems the UM-Pol delta got stuck at the starting position. For both UM and AC, there is no clear stalling at the loop position. Why is that?
Line 347, typographical error: 'Stimulated' should be 'Stimulate'.
Fig 5: Please check the lane labelling. I believe the middle lane has been labelled as all negative by mistake. Thus, there is one label extra. Again, please do explain how the gel was quantified.
In this assay, the accessory proteins are used by themselves and all of them increase the synthesis activity of Pol delta. The surprising part is how is PCNA loading without RFC? And why is RFC increasing Pol delta processivity without PCNA? Please explain.
Reviewer 2 Report
This is a well written biochemistry article describing the impacts of in vitro acetylation of human pol d on its polymerase, strand displacement and secondary structure resolution activities. The authors find that a portion of human POLD1 is acetylated in vivo and that in vitro this acetylation stimulates its various activities. The experiments are generally of good quality and the data looks solid but a lack of information regarding technical replicates and statistical significance of some of the key results needs to be addressed to make the manuscript suitable for publication. Additionally, a number of key controls are missing and are needed to make sure that the observed stimulatory effects of acetylation on pol d activities are real.
Major point. The acetylated pol d is prepared using p300, acetyl-coA and a histone acetyltransferase buffer whereas the unacetylated form is simply incubated in the buffer and then used for the various assays. As it does not appear that p300 is removed from the acetylated pol d preparation prior to the assays, it is possible that it could directly stimulate the polymerase, strand displacement and secondary structure resolution activities of pol d. Some of the key conditions should be redone using pol d incubated with p300 and the buffer but without acetyl-coA as the control condition. If this is how assays were carried out, then it should be clearly stated in the materials and methods section.
Minor points.
1. In figure 1B, a normal rabbit IgG pulldown should be included in the IP experiment to make sure that the immunoprecipitation is specific.
2. In all figure legends, the number of replicates for all experiments should be provided. In panel 2B, statistical significance should be established for the difference in full length synthesis between ac- and unmodified pol d.
3. In figure 5, there are 15 lanes not 16, the labeling should be corrected.
4. On line 380 it is stated that dNTPs were present at 150 uM concentration in figure 6B but in the legend 100 uM is indicated. Which concentration is correct ?
5. Since no in vivo role for pol d acetylation has been described by the authors, the sentence on lines 471-472 should be changed to reflect this.
Typos and grammar
Line 119 and elsewhere in the text : Escherichia coli
Line 130 d
Line 148 EcoRI-XhoI
Line 150 EcoRI-XhoI
Line 216 immunoprecipitates
Line 232 subunit p125 is likely acetylated
Line 245 in vitro
Line 299 its ability to displace this sequence seems to be…
Legend figure 4 quadruplex… there is also a missing part of the last sentence.
Line 346 Can Stimulate Acetylated Polymerase d
Line 350 structured regions [37,38].
Font size in the legend to figure 6.
Line 419 has been displaced
Line 459 especially at the telomeric regions which contain TTAGGG/AATCCC repeats
Line 462 demonstrated that G-quadruplex strctures…
Line 467 confirm that such…
Round 2
Reviewer 2 Report
With the added figures and controls, the manuscript is now suitable for publication. The layout of the article should be corrected as figure 1 is strangely placed.
Typos :
Line 457 : Synthesis is Upon encountering the second (correct this sentence)
Line 665 : suggests that acetylation of pol d
Supp. Fig 2 varying concentrations
Author Response
Reviewer # 2:
We have made the minor edits suggested to the revised manuscript.